# Pro-Apoptotic and Anti-Cancer Activity of the Vernonanthura Nudiflora Hydroethanolic Extract

**DOI:** 10.3390/cancers15051627

**Published:** 2023-03-06

**Authors:** Almog Nadir, Anna Shteinfer-Kuzmine, Swaroop Kumar Pandey, Juan Ortas, Daniel Kerekes, Varda Shoshan-Barmatz

**Affiliations:** 1Department of Life Sciences, Ben-Gurion University of the Negev, Beer-Sheva 84105, Israel; 2National Institute for Biotechnology in the Negev, Ben-Gurion University of the Negev, Beer-Sheva 84105, Israel; 3Sigma Inc., 5701 Dogwood Dr., Orlando, FL 32807, USA

**Keywords:** apoptosis, cancer, metabolism, mitochondria, plant extract, VDAC1

## Abstract

**Simple Summary:**

Natural products derived from plants have numerous clinical applications, including anti-cancer activity. In the present study, we identified three different plant extracts as strong inducers of cell death that were not reported previously. We focused on the most potent of these plants, *Vernonanthura nudiflora* (Vern). We demonstrated that the plant extracts obtained by treatment with a water and ethanol mixture killed tumor cells via multiple routes. These include impairing cell energy and metabolism, generating reactive oxygen species, increasing intracellular Ca^2+^, and inducing mitochondria-mediated apoptosis. We connected these activities to increased levels of the mitochondrial gatekeeper protein, VDAC1, which is associated with metabolism and apoptosis regulation. In a glioblastoma mouse model, Vern extract strongly inhibited tumor growth and induced massive tumor cell death, including cancer stem cells, by inhibiting blood supply and modulating the tumor microenvironment. The multipronged effects of hydroethanolic Vern extract make it a promising candidate for treating cancer.

**Abstract:**

The mitochondrial voltage-dependent anion channel 1 (VDAC1) protein is involved in several essential cancer hallmarks, including energy and metabolism reprogramming and apoptotic cell death evasion. In this study, we demonstrated the ability of hydroethanolic extracts from three different plants, *Vernonanthura nudiflora* (Vern), *Baccharis trimera* (Bac), and *Plantago major* (Pla), to induce cell death. We focused on the most active Vern extract. We demonstrated that it activates multiple pathways that lead to impaired cell energy and metabolism homeostasis, elevated ROS production, increased intracellular Ca^2+^, and mitochondria-mediated apoptosis. The massive cell death generated by this plant extract’s active compounds involves the induction of VDAC1 overexpression and oligomerization and, thereby, apoptosis. Gas chromatography of the hydroethanolic plant extract identified dozens of compounds, including phytol and ethyl linoleate, with the former producing similar effects as the Vern hydroethanolic extract but at 10-fold higher concentrations than those found in the extract. In a xenograft glioblastoma mouse model, both the Vern extract and phytol strongly inhibited tumor growth and cell proliferation and induced massive tumor cell death, including of cancer stem cells, inhibiting angiogenesis and modulating the tumor microenvironment. Taken together, the multiple effects of Vern extract make it a promising potential cancer therapeutic.

## 1. Introduction

Numerous natural products with anti-cancer activity used clinically, such as paclitaxel docetaxel and taxol, are derived from plants [1]. Moreover, some compounds such as resveratrol that are produced in plant species considered to have health benefits are also shown to have pro-apoptotic effects, inducing cell death, and, as such, they act as anti-cancer agents [2,3,4,5]. Similarly, curcumin expresses a variety of therapeutic properties, including antioxidant, anti-inflammatory, and antiseptic activities, as well as anticancer effects in a variety of biological pathways involved in mutagenesis, apoptosis, tumorigenesis, cell cycle regulation, and metastasis [6]. Quercetin, a polyphenol derived from plants, has also been shown to have a wide range of biological actions, including anti-carcinogenic, anti-inflammatory, and antiviral activities, as well as attenuating lipid peroxidation, platelet aggregation, and capillary permeability [7]. In addition to the many plant species that are already used to treat or prevent the development of cancer, several species of plants have demonstrated anti-cancer properties and are used as herbal medicines in developing countries [8].

Here, we focus on the activity of extracts derived from three different plants. The *Vernonanthura nudiflora* species of perennial plant in the family Asteraceae includes more than 23,500 species spread over about 1600 genera [9], with distribution in Argentina, Brazil, and Uruguay [10]. The *Vernonanthura (Vernonia)* genera includes more than 1000 species [11]. The anti-proliferative and antioxidant activities of an organic extract of *Vernonanthura nudiflora* and some of its constituents have already been reported [12]. In addition, some metabolites isolated from the flowers of *Vernonanthura nudiflora* showed antimicrobial activities [9]

The second plant tested for cytotoxicity is the *Baccharis trimera* used in folk medicine for the treatment of gastrointestinal disorders and hepatic diseases [13,14]; other biological activities reported for *B. trimera* include antihepatotoxic, antidiabetic, schistosomicidal, antioxidant, antinociceptive, and anti-inflammatory effects that are attributable to flavonoids, diterpenes, triterpenes, saponins, essential oils, and caffeoylquinic acids [15,16].

The third plant is *Plantago major* from the Plantaginaceae family, commonly known as *great plantain,* and used as a medicinal plant [17]. *Plantago major* contains several active compounds such as flavonoids, polysaccharides, terpenoids, lipids, iridoid glycosides, and caffeic acid derivatives [18]. It is used to treat various diseases such as constipation, cough, wounds, infection, fever, bleeding, and inflammation. In addition, water and ethanol extracts of *Plantago major* leaves show anti-inflammatory activity on oral epithelial cells. 

Here, we present the pro-apoptotic activity of the hydroethanolic extracts from the indicated three plants while deciphering their mode of pro-apoptotic, anti-cancer activity involving mitochondria-mediated apoptosis and the mitochondrial gatekeeper protein, the voltage-dependent channel 1 (VDAC1). 

Mitochondria are central to essential life functions for the generation of cellular energy and critical components of the biosynthetic pathways, and function as points for cellular decisions leading to apoptosis. One of the proteins in control of these cellular life and death decisions is the mitochondrial protein VDAC1. Proper cell activity requires an efficient exchange of molecules between the mitochondria and cytoplasm. Lying in the outer mitochondrial membrane (OMM), VDAC1 assumes a crucial position in the cell, forming the main interface between the mitochondrial and cellular metabolisms [19,20,21]. VDAC1 is a key protein in regulating metabolism, controlling the passage of adenine nucleotides, other metabolites, and Ca^2+^ in and out of mitochondria. VDAC1 is also an essential protein in regulating mitochondria-mediated apoptotic cell death and controls other biological and cellular functions [19,20,21]. VDAC1 is overexpressed in various cancer cell lines and different tumors, pointing to its importance for their survival [20,22,23,24,25]. The crucial role it plays in regulating the metabolic and energetic functions of mitochondria in cancer cells is demonstrated by findings that downregulating VDAC1 expression using siRNA decreases energy production and cell growth and inhibits tumor growth [22,26,27,28,29,30,31]. VDAC1 is overexpressed in many diseases other than cancer, and its overexpression is associated with cell death induction [32]. 

Mitochondria play a central role in apoptosis. During the transduction of an apoptotic signal into the cell, an alteration in the mitochondrial membrane permeability occurs [33,34]. This allows the release of apoptogenic proteins such as cytochrome c (Cyto c), apoptosis-inducing factor (AIF), and second mitochondria-derived activator of caspase/direct inhibitor of apoptosis-binding protein with low pI (Smac/DIABLO) [20,35]. When released from the mitochondria, all participate in the complex processes resulting in the activation of proteases and nucleases, leading to DNA and protein degradation and ultimately to apoptotic cell death [33]. Several mechanisms for releasing the pro-apoptotic proteins have been proposed [19,36,37]. These include a large channel formed by Bax and/or Bak oligomers [38,39] and a channel formed by hetero-oligomers of VDAC1 and Bax [40,41] or VDAC1 oligomers [36,37,42,43,44,45,46,47]. Defects in the regulation of apoptosis are often associated with drug resistance and diseases such as cancer [48], with apoptosis evasion being a cancer hallmark [49].

All the apoptotic proteins known to translocate to the cytoplasm following an apoptotic stimulus reside in the mitochondrial intermembrane space (IMS). Thus, only the permeability of the OMM needs to be modified for their release [50,51,52,53]. Hence, VDAC1 as an OMM channel could mediate Cyto c release. Recently, we demonstrated that VDAC1 oligomers form a large channel that mediates the release of Cyto c and other pro-apoptotic proteins [47,54]. Moreover, we found that cisplatin, selenite, H_2_O_2_, UV light, and more lead to apoptosis by inducing VDAC1 overexpression, thereby shifting the equilibrium towards oligomers, which leads to the release of pro-apoptotic proteins and, subsequently, apoptosis [43,44,45,47,54,55,56,57]. 

In the present study, we demonstrate that hydroethanolic extracts from three different plants, *Vernonanthura nudiflora* (Vern), *Baccharis trimera* (Bac), and *Plantago major* (Pla), can induce apoptotic cell death by increasing VDAC1 expression levels, its oligomerization, and subsequent apoptotic cell death. In addition, the plant extracts increased intracellular Ca^2+^ and ROS production and reduced cell survival. Vern extract and one of its compounds, phytol, inhibited tumor growth and reversed tumor oncogenic properties. 

The plant extracts tested here showed pro-apoptotic anti-cancer activity and thus represent a promising therapeutic candidate for cancer treatment.

## 2. Materials and Methods

### 2.1. Materials

4′,6-diamidino-2-phenylindole (DAPI), dimethyl sulfoxide (DMSO), propidium iodide (PI), Tris, carbonyl cyanide p-trifluoro-methoxyphenyl hydrazone (FCCP), tetramethylrhodamine, methyl ester (TMRM), and trypan blue were purchased from Sigma (St. Louis, MO, USA). Dithiothretol (DTT) was purchased from Thermo Fisher Scientific (Waltham, MA, USA). Annexin-V (FITC) was obtained from Alexis Biochemicals (Lausen, Switzerland). Dulbecco’s modified Eagle’s medium (DMEM) and phosphate-buffered saline (PBS) were purchased from Gibco (Grand Island, NY, USA). Fluo-4-AM and MitoSOX-Red were acquired from Invitrogen (Waltham, MA, USA). TUNEL was obtained from Promega (Madison, WI, USA). Primary and secondary antibodies used in immunoblotting and immunofluorescence (IF), as well as their dilutions, are listed in Appendix A, and XTT cell viability assay kits were obtained from Biological Industries (Beit Haemek, Israel). TLC silica gel 60 F254 plates were obtained from Merck (Darmstadt, Germany). 

### 2.2. Cell Lines and Culture

U-87MG (human glioblastoma), SH-SY5Y (human neuroblastoma), HeLa (human cervix adenocarcinoma), MEFs (mouse embryonic fibroblasts), and PC-3 (human prostate cancer cells) were maintained at 37 °C and 5% CO_2_ in DMEM medium supplemented with 10% FBS, 1 mM L-glutamine, 100 U/mL penicillin, and 100 μg/mL streptomycin. Mycoplasma contamination was routinely evaluated on cell lines. 

### 2.3. Plant Extracts

*Vernonanthura nudiflora, Baccharis trimera*, and the *Plantago major* plants were washed with distilled water, and the aerial parts were naturally dried up to a 50% reduction in total mass. The material was introduced into reactors for maceration, with 0.20 g botanic material per 1 mL and 70% of a hydroalcoholic solution of ethanol/water 70/30, and agitated for a period of 21 days. The extract was later filtered through a filter with a pore size of 15 μm and kept at 4 °C. Before use, the hydroalcoholic was centrifuged for 5 min at 15,000× *g*.

The mixture was composed of a hydroethanolic filtered extract of *Vernonanthura nudiflora* (40%), *Baccharis trimera* (40%), and *Plantago major* (20%). 

### 2.4. Cell Treatment with Plant Extracts and Cell Death Analysis

Cells (6 × 10^5^/mL at 70–80% confluence) were incubated with ethanol extract from Vern, Bac, or Pla plant extracts or their mixture at the indicated dilution in 2000 µL culture medium for 24 h or the indicated time at 37 °C and 5% CO_2_. The cells were then trypsinized, centrifuged (1500× *g*, 5 min), washed with PBS, and analyzed for the desired activity. For cell death analysis, propidium iodide (PI) staining was performed by adding PI (6.25 µg/mL) to the cells, followed by immediate analysis by flow cytometry with the iCyt sEC800 -flow cytometry analyzer (Sony Biotechnology Inc., San Jose, CA, USA) and analysis with EC800 software.

For PI and annexin V-FITC staining, cells (2 × 10^5^), untreated or treated with the plant extracts, were collected (1500× *g* for 5 min), washed, and resuspended in 200 μL binding buffer (10 mM HEPES-NaOH, pH 7.4, 140 mM NaCl, and 2.5 mM CaCl_2_). Annexin V–FITC staining was performed according to the recommended protocol. Cells were then washed once with binding buffer and resuspended in 200 μL binding buffer, to which PI was added immediately before flow cytometric analysis by flow cytometry with the iCyt sEC800 -flow cytometry analyzer (Sony Biotechnology Inc., San Jose, CA, USA) and analysis with EC800 software. At least 10,000 events were collected and recorded on a dot plot. 

### 2.5. Cell Viability Assay

The effect of the plant extracts on SH-SY5Y cell survival was assayed using an XTT-based kit (Biological Industries, Beit Haemek, Israel) according to the manufacturer’s protocol. Cells were seeded in a 96-well plate and incubated at 37 °C with 5% CO_2_, and 24 h later were treated with different concentrations of the extracts for the time indicated in the figure legends. XTT reagent was added to each well, and the absorbance was measured at 450 nm and 630 nm (Tecan, Infinite M1000, Mannedorf, Switzerland). The absorbance obtained at 630 nm was subtracted from the absorbance at 450 nm to obtain the specific reduced XTT reaction product. 

### 2.6. Determination of Reactive Oxygen Species (ROS), Mitochondria Membrane Potential, and Intracellular Ca^2+^ Levels 

Mitochondrial ROS Determination: For measuring mitochondrial accumulated ROS, SH-SY5Y cells were seeded in a 6-well plate (1 × 10^5^/well). Cells were treated for 24 h with the indicated plant extract, and then were incubated with MitoSOX-Red, a mitochondrial superoxide indicator for live-cell imaging, for 10 min at 37 °C. Fluorescence was measured using flow cytometry (iCyt, Sony Biotechnology, San Jose, CA, USA). At least 10,000 events were recorded on the FL2 detector, represented as a histogram, and analyzed with EC800 software (Sony Biotechnology, San Jose, CA, USA). Positive cells showed a shift to an enhanced level of green fluorescence (FL2).

Mitochondrial Membrane Potential Determination: Mitochondrial membrane potential was determined using TMRM, a potentially sensitive dye, and a plate reader. SH-SY5Y cells were treated for 24 h with the indicated Vern plant extract and subsequently incubated with TMRM (400 nM, 20 min). The cells were then washed twice with PBS and examined withby flow cytometry with the iCyt sy3200 Benchtop Cell Sorter/Analyzer (Sony Biotechnology Inc., San Jose, CA, USA) and analysis with EC800 software. FCCP-mediated dissipation served as the control.

Cytosolic Ca^2+^ levels [Ca^2+^]i measurements: [Ca^2+^]i was analyzed using Fluo-4-AM. Cells were harvested after the appropriate treatment, collected (1500× *g* for 10 min), washed with HBSS buffer (5.33 mM KCl, 0.44 mM KH_2_PO_4_, 138 mM NaCl, 4 mM NaHCO_3_, 0.3 mM Na_2_HPO_4_, 5.6 mM glucose), supplemented with 1.8 mM CaCl_2_ (HBSS+), and incubated with 2 μM Fluo-4 in 200 μL of HBSS(+) buffer in the dark for 20 min at 37 °C. After washing the remaining dye, the cells were incubated with 200 μL HBSS(+) buffer, and changes in [Ca^2+^]i were measured immediately by FACS and analyzed by flow cytometry with the iCyt sy3200 Benchtop Cell Sorter/Analyzer (Sony Biotechnology Inc., San Jose, CA, USA) and analysis with EC800 software. Positive cells showed a shift to an enhanced level of green fluorescence (FL1). 

Changes in cellular Ca^2+^ were monitored in live cells using the high content Operetta screening system (Perkin-Elmer, Hamburg, Germany). In each well, ten fields were imaged using a 20× wide field objective with an excitation filter of 520–550 nm and emission filter of 560–630 nm.

### 2.7. Cross-Linking Experiments 

Cells were treated with the plant extract for the indicated time and concentration, harvested, washed with PBS, pH 8.3, and incubated for 15 min with the cross-linking reagent EGS at a ratio of 1 mg protein/mL/100 μM EGS. Aliquots (30 µg protein) were subjected to SDS-PAGE and immunoblotting using anti-VDAC1 antibodies. Quantitative analysis of VDAC1 dimers was performed using FUSION-FX (Vilber Lourmat, Marne-la-Vallée, France).

### 2.8. Gel Electrophoresis and Immunoblotting

Cells or tumor tissues were lysed using lysis buffer (50 mM Tris-HCl, pH 7.5, 150 mM NaCl, 1 mM EDTA, 1.5 mM MgCl_2_, 10% glycerol, 1% Triton X-100, supplemented with a protease inhibitor cocktail (Calbiochem, Welwyn Garden City, UK)). The lysates were then centrifuged at 12,000× *g* (10 min at 4 °C), and protein concentration was determined. Aliquots (10–20 μg of protein) were subjected to SDS-PAGE and immunoblotting using various primary antibodies (sources and dilutions are provided in Appendix A), followed by incubation with appropriate HRP-conjugated secondary antibodies (i.e., anti-mouse, anti-rabbit). Blots were developed using enhanced chemiluminescence (Biological Industries, Beit Haemek, Israel). Band intensities were analyzed by densitometry using FUSION-FX (Vilber Lourmat, Marne La Vallée, France) software, and values were normalized to the intensities of the appropriate β-actin signal that served as a loading control.

### 2.9. Gas Chromatography–Mass Spectroscopy (GC-MS) Analysis

CG-MS analysis was carried out using a 7890B Mass-Detector; 5977A, Agilent Technologies; Column 5MS UI. The compounds were identified using Library Name W 10N 14L (NIST MS Search 2.2). The various names representing each compound, quality of identification (maximum is 100%), and peak area (Ab*s) are given in Appendix A.

### 2.10. TLC Separation 

TLC silica gel 60 F254 plates (Merck, 20 × 20 cm) were used to separate the ethanol plant extracts and phytol and ethyl linoleate, using a mobile phase mixture of petroleum ether:diethyl ether:acetic acid (85:15:1 V/V/V). The plates were air dried and visualized by exposure to iodine vapor.

### 2.11. Xenograft Mouse Model 

U-87MG cells (1.8 × 10^6^ cells/mouse) were inoculated subcutaneously (s.c.) into the hind leg flanks of athymic eight-week-old male nude mice (Envigo). Tumor size was measured using a digital caliper, and volume was calculated. When it reached 50 mm^3^, mice were randomly divided into several groups (5 mice/group). One group was intratumorally injected with PBS containing 5% ethanol (final concentration in the tumor was 0.14%), and other groups were treated with Vern plant extract to a final dilution of 1:100, 1:250, 1:300, or 1:500 or with phytol to a final concentration of 75 μM. The xenografts were injected three times a week. The mice were sacrificed 34 days post-cell inoculation, and tumors were excised. Tumors were fixed in 4% buffered formaldehyde, paraffin-embedded, and processed for immunofluorescence (IF). These experimental protocols were approved by the Institutional Animal Care and Use Committee of Ben-Gurion University.

### 2.12. Immunofluorescence (IF) of Tumor Tissue Sections

Formalin-fixed, paraffin-embedded sections (5 μm thick) of U-87MG cell-derived tumors from control and Vern plant extract- or phytol-treated tumors were deparaffinized by placing the slides at 60 °C for 1 h and using xylene, followed by rehydration with a graded ethanol series (100–50%). Antigen retrieval was performed in 0.01 M citrate buffer (pH 6.0) at 95 °C–98 °C for 20 min. After washing sections in PBS, pH 7.4, sections were incubated in 10% normal goat serum, 1% BSA in PBS containing 0.1% Triton X100 for 2 h, followed by overnight incubation at 4 °C with primary antibodies (Appendix A). Sections were washed thoroughly with PBS, pH 7.4, containing 0.1% Triton-X100 (PBST), incubated with the fluorescently labeled secondary antibodies (Appendix A) for 2 h, washed five times with PBST, and cover-slipped with fluoroshield mounting medium (Immunobioscience, Mukilteo, WA, USA). Fluorescent images were viewed with an Olympus IX81 confocal microscope. Quantitation of protein levels, as reflected in the staining intensity, was analyzed in the whole area of the sections using Image J software.

### 2.13. TUNEL Assay

Paraffin-embedded fixed tumor sections (5 μm thick) were processed for a Terminal deoxynucleotidyl transferase (TdT)-mediated dUTP nick-end labeling TUNEL assay using the Dead End Fluorometric TUNEL system according to the manufacturer’s instructions. Sections were deparaffinized, equilibrated in PBS, permeabilized with proteinase K (20 μg/mL in PBS), post-fixed in 4% paraformaldehyde, and incubated in TdT reaction mix for 1 h at 37 °C in the dark. Slides were then washed in saline–sodium citrate buffer, counter-stained with PI (1 µg/mL), and cover slipped with fluoroshield mounting medium. Fluorescent images of apoptotic cells (green) and cell nuclei (red) were captured using a confocal microscope (Olympus 1 × 81). Quantification analysis of stained slides was performed using an Image J program.

### 2.14. Statistics and Data Analysis 

Means ± SE of results obtained from three independent experiments are presented. Statistical significance is reported at *p* < 0.05 (*), *p* < 0.01 (**), *p* < 0.001 (***), or *p* < 0.0001 (****).

## 3. Results 

### 3.1. Apoptosis Induction by the Hydroethanolic Plant Extracts 

The cell death activity of the three plant extracts (Vern, Bac, Pla) alone and combined was analyzed by propidium iodide (PI) staining and flow cytometry analysis (Figure 1A,B). The results show that Vern extract was the most potent, triggering massive cell death, followed by the Pla extract and then the Bac extract. Vern extract (1:500) induced more cell death than all three extracts combined at a 1:100 dilution and Bac extract alone (1:166), suggesting that it is five- and threefold more active, respectively (Figure 1B). 

To determine whether the observed cell death induced by Vern was apoptosis, we analyzed apoptosis by Annexin V/PI and via FACS (Figure 1C and Appendix A). The results show that Vern extract induced apoptotic cell death. 

Next, we analyzed the effects of the plant extract on cell survival using the XTT assay (Figure 1D–F). The Vern extract reduced cell survival following 24, 48, and 72 h incubation. In contrast, Bac extract showed some decreased survival following incubation for 48 h and 72 h, and Pla extract showed no decrease in cell survival. Considering that the XTT assay is based on reduced levels of NADH produced in the mitochondria, these results suggest that Vern plant extract, but not Bac and Pla, induced mitochondrial dysfunction. In addition, the results suggest that the cell death caused by the three plant extracts involves different active compounds and modes of action.

Vern plant extract similar to SH-SY5Y cells induced cell death in other cancer cell lines such as Hela and PC-3 (Appendix A). In contrast, non-cancerous cell lines, such as MEFs, were less sensitive to the Vern plant extract (Appendix A).

The following experiments were conducted to reveal the extract active compounds’ possible modes of action.

The ethanolic plant extracts induced VDAC1 overexpression and oligomerization. 

We have shown that many apoptosis triggers, such as chemotherapy drugs, stress, and radiation, induce VDAC1 overexpression and oligomerization. We suggest that this is a general mechanism common to numerous apoptosis stimuli, although they act via different initiating cascades [42,43,44,46,56]. 

Thus, we tested the effects of the plant extracts on VDAC1 expression levels and its oligomerization (Figure 2). The most active Vern plant extract induced VDAC1 overexpression in both cell lines tested: the neuroblastoma-derived cell line SH-SY5Y and glioblastoma-derived U-87MG cell line (Figure 2A). In both cell lines, Vern extract highly increased the expressed VDAC1 level by three- to fourfold (Figure 2A) and induced similar pro-apoptotic activity (IC_50_ = 1:800) (Figure 2B).

The three extracts, in a concentration-dependent manner, increased VDAC1 oligomeric forms as stabilized by chemical cross-linking using EGS and monitored by immunoblotting [43] (Figure 2C). However, the highest level of VDAC1 oligomerization was induced by the Vern extract (Figure 2C,D), which aligns with the superior potency of its cell death-inducing activity.

Interestingly, we observed the presence of VDAC1 oligomers even without chemical cross-linking, even after exposing the cells to a high concentration (1%) of the detergent SDS and heating at 70 °C for 5 min (Figure 2E,F). The level of oligomeric VDAC1 was highest with Vern extract, as found for VDAC1 overexpression, oligomerization, and apoptosis induction. This suggests that the VDAC1 oligomers induced by the plant extract are very stable. 

The results support the suggestion that the active compounds in the Vern plant extract, via enhancing VDAC1 expression levels, lead to VDAC1 oligomerization and apoptosis.

The active compounds in Vern extract are resistant to high temperatures, as heating the extract for 10 min at 40, 60, or 80 °C had no effect on the extract cell death activity (Appendix A).

### 3.2. Vern Extract Increased Intracellular Ca^2+^ and ROS Production 

Reactive oxygen species (ROS) were shown to induce apoptosis [58]. Thus, we measured mitochondrial ROS and found that cell treatment with Vern extract induced their production (Figure 3A). This increase in mitochondrial ROS suggests that Vern extract at high concentrations induces dysfunction of the mitochondria, as also reflected in the decrease in XTT reduction. 

Several studies have shown that an increase in [Ca^2+^] is involved in apoptosis induction and that Ca^2+^ is required for apoptosis-stimuli-induced VDAC1 overexpression and VDAC1 oligomerization [45,46,59]. Vern extract’s effect on cellular [Ca^2+^] levels was analyzed using Fluo-4 and FACS or by Operetta (Figure 3B–D). Both assays demonstrated that at high levels, this extract increased cellular [Ca^2+^] levels.

Cell treatment with Vern plant extract reduced the mitochondrial membrane potential (Δ**ψ**) only when high (over 80%) cell death was obtained (Appendix A). 

The findings that the increase in ROS production, [Ca^2+^] levels, and dissipation of (Δ**ψ**) were not correlated with cell death and were observed only at high concentrations of the Vern plant extract and over 80% cell death suggest that these effects are due to cell destruction, including of the mitochondria.

### 3.3. GS-MS Analysis of Extracts from Plants Vern, Bac, and Pla

To identify some of the chemical compounds present in the hydroethanolic extracts from the three different plants, Vern, Bac, and Pla, the extracts were subjected to gas chromatography–mass spectroscopy (GC-MS) analysis. Considering only those compounds with a score of over 90% certainty, 12 and 13 compounds were identified in Vern and Pla extracts, respectively, and 20 in the Bac extract.

Some of these compounds are common to the three plant extracts, while others are only two or unique to just one (Appendix A). As expected for ethanol extract, all identified compounds are hydrophobic, containing fatty acid derivatives such as palmitic acid ethyl ester (hexadecanoic acid ethyl ester, linolenic acid ethyl ester, (9,12,15-octadecatrienoic acid ethyl ester), and stearic acid ethyl ester (octadecanoic acid, ethyl ester). We tested the cell death induction activity of two compounds found in the extracts, namely phytol and ethyl linoleate (Appendix A), both commercially available. The relative amounts of phytol and ethyl linoleate in the plant extracts were determined using known quantities of the two compounds and TLC (Figure 4A,B). The amounts of phytol were about 1800, 800, and 1200 nmol/mL (μM) in Vern, Bac, and Pla extracts, respectively.

Next, the activity of the phytol and ethyl linoleate in inducing cell death was analyzed by incubating SH-SY5Y cells with different concentrations (50–200 μM) for 24 or 48 h (Figure 4C). Phytol induced cell death with half-maximal cell death (IC_50_) of about 80% at 70 μM. Ethyl linoleate showed weak cell death activity, increasing from 15% in non-treated cells to about 40% at 200 μM of ethyl linoleate (Figure 4C). Phytol also highly reduced cell survival, as analyzed using the XTT assay (Figure 4D), suggesting that its effect involves mitochondria dysfunction.

Next, we tested whether phytol- and ethyl linoleate-induced cell death was associated with increased VDAC1 expression levels and oligomerization. Both compounds, at the high concentrations used, induced VDAC1 overexpression (Figure 5A,B) and VDAC1 oligomerization (Figure 5C,D) in a concentration-dependent manner. In correlation with the higher activity of phytol in cell death induction, phytol increased both VDAC1 overexpression and oligomerization. 

Since Vern extract induced cell death at a dilution of 500–1000, with phytol concentrations in these dilutions of 1.8 to 3.6 μM, cell death was observed at over 50 μM of phytol, suggesting that other compounds in the plant extracts are involved in cell death induction.

### 3.4. Anti-Tumor Activity of Vern Extract and Phytol 

Next, we tested the effect of Vern extract at two dilutions and phytol on tumor growth (Figure 6). U-87MG cells were inoculated subcutaneously (s.c.) into the hind leg flanks of 7-week-old male athymic nude mice. When the tumor volume was around 50 mm^3^, the mice were divided into four groups with a similar average volume and treated with Vern extract or phytol (Figure 6A). Control tumors were injected with PBS containing 5% ethanol (final concentration in the tumor 0.14%). Groups 2 and 3 were treated with Vern extract at a final dilution in the tumor of 1:250 or 1:500, and Group 4 was treated with phytol (75 μM). Treatment was given three times a week, and tumor growth was monitored (Figure 6B,C). All mice were sacrificed 34 days post-cell inoculation; tumors were excised, weighed (Figure 6D), and fixed; and sections were immunofluorescent-stained for selected proteins.

The results show that the tumors in the control grew exponentially with time and in a similar way when injected with Vern extract to a final dilution of 1:250. On the other hand, tumors treated with a higher dilution of Vern extract, 1:500, showed about a 70% decrease in tumor volume and weight (Figure 6B–D). The results indicate that Vern extract at a higher concentration (1:250) is less effective than at the dilution of 1:500. Similar results with no effect on tumor growth were obtained using a 1:100 dilution of Vern extract (Appendix A). The decreased anti-cancer effect with increased Vern extract concentration may result from the protective activity of other compounds in the extract. 

The results also show that phytol at the concentration used (75 μM) significantly inhibited tumor growth, yet less than Vern extract at 1:500 dilution (Figure 6B–D). Phytol has been shown to induce apoptosis in cells in culture [60,61,62], and, as also shown here, it also decreased cell survival (Figure 4C,D). For the first time, it induced VDAC1 overexpression and VDAC1 oligomerization (Figure 5) and inhibited tumor growth (Figure 6). 

Tumor-fixed paraffin-embedded sections were stained for Ki-67, a proliferation marker, showing that both Vern extract and phytol inhibited cell proliferation by about 80% (Figure 6E,F). Finally, we analyzed apoptosis by TUNEL staining (Figure 6G,H). While no TUNEL-positive cells were apparent in control tumors, most of the cells were TUNEL-positive in the Vern extract-treated tumors and to a lesser extent in phytol-treated tumors, with staining co-localizing with PI nuclear staining (Figure 6G, white arrows). Thus, Vern extract was more effective than phytol in cell death induction. 

The results indicate that the treatments induced apoptotic cell death and suggest that the marked decrease in tumor size in the Vern extract- and phytol-treated xenografts can be attributed to both inhibition of cell proliferation (decreased Ki-67 staining) and cell death induction. 

Next, we analyzed the effect of tumor treatment with Vern extract or phytol on the expression levels of proteins associated with metabolism, the microenvironment, and cancer stem cells (Figure 7, Figure 8 and Figure 9).

### 3.5. Vern Extract and Phytol Reduced the Expression of the Metabolic Enzyme in a Xenograft Mouse Model 

The metabolic alterations that occur during malignant transformation involve a spectrum of functional aberrations and mutations that contribute to elevated glycolysis and increased expression levels of glucose transporters (Glut-1) and glycolytic enzymes as hexokinase (HK-I) [63]. IF of the control tumors derived from U-87MG cells showed high expression levels of Glut-1 and glyceraldehyde three-phosphate dehydrogenase (GAPDH) that were decreased in tumors treated with Vern extract or phytol (Figure 7A,B). Similarly, the expression of VDAC1 and HK-I was reduced in the Vern extract- and phytol-treated tumors (Figure 7C,D). The decreased expression levels of metabolism-related enzymes in the Vern extract- and phytol-treated tumors suggest decreased energy production in these treated tumors.

### 3.6. Vern Extract and Phytol Modulate the Tumor Microenvironment 

A tumor contains cancer cells and non-cancerous cells, creating the tumor microenvironment (TME). Besides cancer cells, a tumor has fibroblasts [64], immune system cells [65], blood vessels [66], and extracellular matrix (ECM) components [67]. The TME plays a vital role in cancer growth and spread.

Angiogenesis is an underlying promoter of tumor growth, invasion, and metastases, with glioblastoma (GBM) being highly angiogenic [68]. Immunostaining of endothelial cell marker CD-31 showed that in the Vern extract- or phytol-treated tumors, there was a significant decrease in the number of blood vessels, with quantitation revealing decreases of about 70% and 60% in Vern extract and phytol-treated tumors, respectively, relative to control tumors (Figure 8A,B).

The effects of Vern extract and phytol on the TME were analyzed by IF staining for the fibroblast marker alpha-smooth muscle actin (α-SMA) (Figure 8C,D). Tumors treated with either Vern extract or phytol showed decreased α-SMA expression. 

Accumulated recent evidence supports the cancer stem cell (CSC) hypothesis, which suggests that a sub-population of malignant cells exhibit the stem cell properties of self-renewal and differentiation. CSCs are resistant to conventional cytotoxic/anti-proliferative therapies. In GBM, the proteins Sox2, CD133, SSEA1, CD49f, Musashi-1, and Nestin are considered to be glioma stem cell CSC markers. The IF staining for Sox2 and Nestin of the tumor specimens demonstrated that in tumors treated with plant Vern extract or phytol, the expression levels of these CSC markers was highly reduced, by about 70% (Figure 9). These results indicate that Vern plant extract and phytol treatment of U-87MG-derived tumors eliminated CSCs associated with tumor recurrence.

## 4. Discussion

Recently, significant attention has been placed on using nutraceuticals as therapeutic agents inducing cell death and suppressing cancer growth as an alternative treatment for cancer or in combination with chemotherapy [2,3,4,5,69,70,71]. Screening for plant-derived compounds with anti-neoplastic activity has contributed to identifying resveratrol, quercetin, curcumin, and others. In addition to their anti-cancer activity, these compounds also showed antioxidant, anti-inflammatory, anti-viral, and neuroprotective properties, lowered blood pressure, and improved cardio-metabolic markers and anti-aging effects [72,73,74,75,76]. 

Here, we present the activity of hydroethanolic extracts from three different plants—*Vernonanthura nudiflora*, *Baccharis trimera*, *Plantago major*, and their mixture—in cell death induction, VDAC1 overexpression, and oligomerization. The effects of Vern plant extract and one of its constituents, phytol, on tumor growth and oncological properties were tested. 

### 4.1. Plant Extracts Inducing Cell Death Involve VDAC1 Overexpression and Oligomerization

Recently, VDAC1 has been recognized as a regulator of mitochondria-mediated apoptosis [36,37,42,43,44,45,46,47,54,55,56,57]. We demonstrated that many apoptosis inducers lead to VDAC1 overexpression and its oligomerization, resulting in the formation of a large channel that enables the release of pro-apoptotic protein from the IMS to the cytosol, thereby activating apoptosis [42,47,57]. 

This study showed that the three plant extracts induced massive cell death at relatively low doses (1:1000 of the original ethanolic extract). The plant extracts’ active compounds inducing apoptosis involved increased VDAC1 expression levels and oligomerization and, thereby, apoptosis. Thus, we propose that phytol and the plant extracts’ activation of apoptosis involve overexpression of VDAC1, shifting the equilibrium towards VDAC1 oligomers, allowing Cyto *c* release, and thereby, apoptosis [42,47,57] (Figure 10). 

We have proposed a model for plant extract and phytol inducing VDAC1 overexpression leading to VDAC1 oligomerization, forming a large channel, mediating the release of apoptogenic proteins from the intermembrane space (IMS) to the cytosol, and activating the apoptosis cascade. 

The active compound(s) in the plant extracts inducing VDAC1 overexpression may involve several possible mechanisms, such as the increase in ROS and intracellular Ca^2+^ levels, as both were shown to regulate gene expression. Ca^2+^-dependent gene transcription has been demonstrated in neurons [77,78,79] and other cells [80,81]. ROS was shown to upregulate gene expression for death receptors such as TRAIL (TNF-related apoptosis-inducing ligand) that appear to be mediated by transcription factors such as CHOP (C/EBP homologous protein) and p53 [82,83].

Here, we demonstrated that Vern extract highly elevated intracellular Ca^2+^ levels, as monitored using Fluo-4 and FACS analysis or by cell imaging using the Operetta imaging system (Figure 3). Similarly, Vern extract induced ROS production (Figure 3). Thus, these findings suggest the involvement of Ca^2+^ and ROS in triggering transcription factors controlling VDAC1 expression, with the increased VDAC1 level leading to its oligomerization, and apoptotic cell death [42,43,47,54]. The active molecule(s) in the extracts responsible for VDAC1 overexpression are yet to be identified.

### 4.2. GC-MS Analysis of the Ethanolic Plant Extracts

A preliminary GC-MS study showed that the ethanolic plant extracts contain many chemical entities at different levels (Appendix A). The major compounds are fatty acids in their ethyl ester forms, such as palmitic acid ethyl ester (hexadecanoic acid ethyl ester), linolenic acid ethyl ester (9,12,15-octadecatrienoic acid ethyl ester), and stearic acid ethyl ester (octadecanoic acid ethyl ester). Interestingly, analysis of the constituents of ethanol root extract of the plant Rauwolfia vomitoria using GC-MS also showed the presence of fatty acids. Still, these were different from those found in the plant extracts tested here, such as ethyl oleate (10.59%), 9,12-octadecadienoic acid ethyl ester (8.26%), and palmitic acid, also known as hexadecanoic acid ethyl ester (8.11%) [84]. 

Here, we showed that one of the identified compounds in the ethanolic plant extracts, phytol, was found to induce cell death and VDAC1 overexpression and oligomerization. Phytol has been reported to induce both apoptosis and protective autophagy [85]. 

Quantitative analysis of phytol amounts in the three plant extracts indicates that its concentration was highest in the Vern extract (5.9 mM) and about 7 mM and 2 mM in extracts B and C, respectively. Phytol-induced cell death was obtained at high concentrations (50 to 200 μM). Based on the estimated phytol concentration in the plant extracts and that Vern extract induced cell death at a 1000-fold dilution, the phytol concentration is about 13.5 μM below its effective concentrations in cell death induction (Figure 4). Thus, it is most likely that other compounds in the extracts contribute to its biological activity; however, their identification is the topic of another study.

Interestingly, our findings that phytol induced VDAC1 overexpression and oligomerization agree with the results that phytol was shown to modulate transcription in cells via a transcription factor—the peroxisome proliferator-activated receptor alpha (PPAR)—involved in regulating lipid metabolism in various tissues [86,87]. Phytol directly activates PPAR-alpha and regulates gene expression involved in lipid metabolism in PPAR-alpha-expressing HepG2 hepatocytes. It also modulates the retinoid X receptors (RXRs), which are nuclear receptors activated by various endogenous and natural ligands such as 9-cis retinoic acid, n-3 polyunsaturated fatty acids, and phytanic acid [88]. Thus, phytol may enhance VDAC1 expression by modulating transcription factor(s). 

### 4.3. Vern Extract and Phytol Inhibit Tumor Growth and Alter Tumor Oncogenic Properties

The effects of Vern extract and phytol on tumor growth and tumor oncogenic properties were tested using a U-87MG-cell-derived tumor-based GBM mouse model. GBM is an aggressive brain cancer with high rates of relapse and mortality, mutational diversity, and poor treatment options. 

The Vern extract decreased tumor size by about 70% when used at the high dilution of 1:500 but was less effective at higher amounts, such as at 1:200 or 1:100 dilutions (Figure 6 and Appendix A). This finding can be explained when considering the composition of the extract compounds revealed by both GS-MS and LC-MS/MS analyses. In addition to the cell death-induced compounds, Vern plant extract contains compounds that support cell growth and are protective against cell death (Appendix A). We suggest that the pro-survival compounds are active at high concentrations and overcome the activity of the pro-cell death compounds. At high dilution, the levels of pro-survival compounds are below their active concentrations; thereby, the effects of the pro-apoptotic compounds, which act at low concentrations, are evident. This observation suggests that it is possible to control the desired activity, supportive/anti- or pro-cell death, according to the extract concentration. 

The results also show that phytol at the concentration used (75 μM) inhibited tumor growth. Phytol in cells in culture has been shown to induce apoptosis and protective autophagy [85]. Here, for the first time, it was demonstrated to inhibit tumor growth, and as discussed below, similar to the Vern extract, it induced multiple effects on the tumor. The impact of Vern extract on the tumors cannot be due to phytol, as at a dilution of 1:500, it contains about 30 μM phytol, which induced only 10% cell death (Figure 4C). 

The inhibition of tumor growth involves both inhibition of cell proliferation, reflected in an 80% decrease in the levels of the cell proliferation marker, Ki-67 (Figure 6E,F), and cell death induction, showing that the Vern extract is about fivefold more active than phytol (Figure 6G,H).

Tumors require changes in the cellular metabolism and bioenergy of cancer cells [22], and their metabolic adaptation provides the tumor with the precursor needed for the biosynthesis of nucleic acids, fatty acids, cholesterol, and porphyrins [63,89]. Mitochondrial metabolism plays a vital role in the survival and development of cancer cells [90]. Here, we demonstrated that both Vern extract and phytol treatment of GBM in a mouse model significantly decreased the expression of metabolism-related enzymes involved in glycolysis and the TCA cycle (Figure 7), leading to reduced cell function and survival. 

The decrease in the expression of metabolism-related enzymes in the tumors treated with Vern plant extract or phytol for 27 days may result from the massive cell death leading to cell distraction, including in the mitochondria and in the degradation of many cell proteins (Figure 7). Similar results were obtained with the VDAC1-based peptide [91].

Vern extract and phytol tumor treatment altered the tumor microenvironment, disrupting tumor–host interactions. The treated tumors showed a reduced expression of angiogenesis markers such as CD-31, decreasing blood supply. 

Chronic inflammation caused by cancer cells stimulates surrounding cells, including fibroblasts and activated fibroblasts, with α-SMA expression producing an extracellular matrix including collagen. The fact that Vern extract and phytol alter the tumor microenvironment is reflected in the decreased α-SMA and Sirius red staining (Figure 8). α-SMA produced by cancer-associated fibroblasts (CAFs) contributes to remodeling and reconstitution to promote invasion and metastasis via the extracellular matrix, growth factors, and protease production [92], as well as to metastasis, and poorer prognosis [93] was highly decreased; thus, the treatment reduced these tumor properties. 

As tumorigenesis is considered an interplay between tumor cells and the surrounding stroma host cells [49,94], alteration in the tumor microenvironment by the treatments suggests that this affects cancer progression, invasiveness, and treatment response. 

Finally, CSCs, with their ability to self-regenerate, are considered to be responsible for initiating tumor growth and recurrence after therapeutic interventions and are associated with tumor resistance to anti-cancer therapies [95]. Here, we showed that tumor treatment with Vern extract or phytol resulted in the elimination of CSCs, as indicated by the decreased expression of the specific markers Nestin and Sox2 (Figure 9) [96]. 

## 5. Conclusions

We found that Vern extract at a high dilution and one of its compounds, phytol, have various effects on tumors, with their anti-cancer effects involving (i) apoptosis induction, (ii) inhibition of cell proliferation, (iii) re-modulation of the tumor microenvironment, (iv) impairment of cancer cell metabolism, and (v) eliminating CSCs, all leading to the observed inhibition of tumor growth. These findings, and considering that the side effects of these plant extracts are minor relative to those of conventional chemotherapy, suggest that the plant extract or a combination of its active compounds (yet to be identified) are a promising therapeutic approach for GBM and various other cancers. 

## Figures and Tables

**Figure 1 cancers-15-01627-f001:**
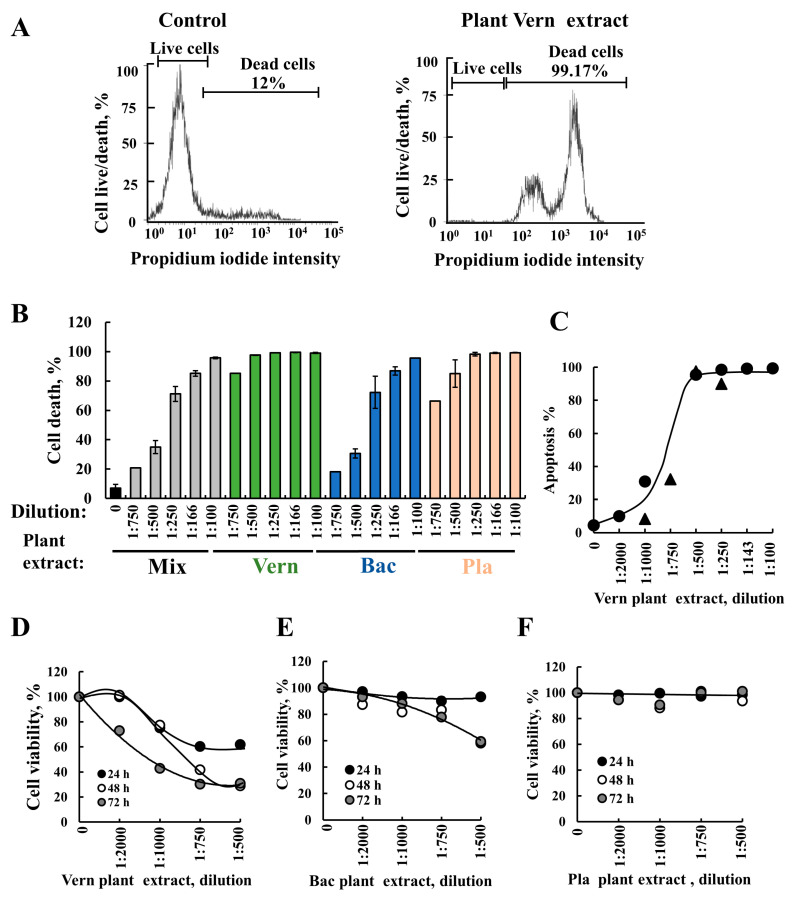
**Plant extracts derived from *Vernonanthura nudiflora* (Vern), *Baccharis trimera* (Bac), and *Plantago major* (Pla) inducing apoptotic cell death**. Cancer cells, SH-SY5Y, were incubated (24 h) with the indicated dilutions of the ethanolic plant extracts: Vern (Vernonanthura nudiflora), Bac (Baccharis trimera), Pla (Plantago major), or their mixture (Vern+Bac+Pla). (**A**) A representative FACS analysis of propidium iodine (PI)-stained cells showing live and dead cells in the control and cells incubated for 24 h with Vern plant extract (1:500). (**B**) Apoptosis was analyzed in cells incubated with the indicated different dilutions of extract from plant Vern, Bac, or Pla extracts, and death cells were analyzed as in (**A**). The results are the mean ± SEM of three independent experiments. (**C**) SH-SY5Y cells were incubated (24 h) with the indicated dilution of Vern plant extract, and then analyzed for apoptosis using Annexin V/PI staining and FACS. (●) and (▲) represent results from two different experiments. (**D**–**F**) SH-SY5Y cell survival as revealed by the XTT assay of cells incubated with different dilutions of the indicated plant extract for 24, 48, or 72 h.

**Figure 2 cancers-15-01627-f002:**
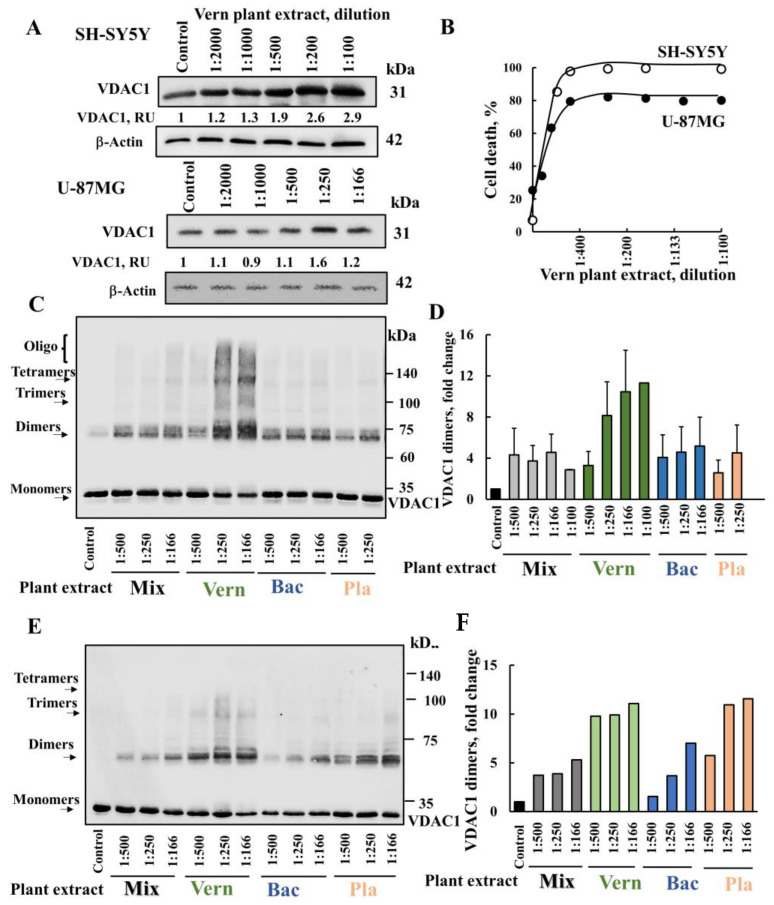
**Vern plant extract induced VDAC1 overexpression, oligomerization, and cell death**. (**A,B**) SH-SY5Y or U-87MG cells were incubated for 24 h with the indicated dilution of Vern plant extract, and then analyzed for VDAC1 expression levels (**A**) by immunoblotting using anti-VDAC1-specific antibodies. Immunoblotting with β-actin as a loading control is also shown. The levels of VDAC1 are given below the blot as relative units (RUs). Samples were also analyzed for cell death (**B**). (**C**,**D**) SH-SY5Y cells (1 mg/mL) were incubated for 24 h with the indicated dilutions of extracts Vern, Bac, Pla, or their mixture and then analyzed for VDAC1 oligomerization by incubation with the cross-linking reagent EGS (100 μM), followed by immunoblotting using anti-VDAC1 antibodies (**C**). The positions of the VDAC1 monomers, dimers, trimers, tetramers, and higher oligomers are indicated. The level of VDAC1 dimers was analyzed using FUSION-FX software (**D**) and presented relative to its levels in control cells subjected to EGS. (**E**,**F**) SH-SY5Y cells were incubated for 24 h with the indicated dilutions of plant extracts Vern, Bac, Pla, or their mixture without EGS treatment, and subjected to immunoblotting. The position of the VDAC1 monomers, dimers, trimers, and tetramers is indicated. The level of VDAC1 dimers was analyzed using FUSION-FX software and presented relative to its levels in control cells (**F**).

**Figure 3 cancers-15-01627-f003:**
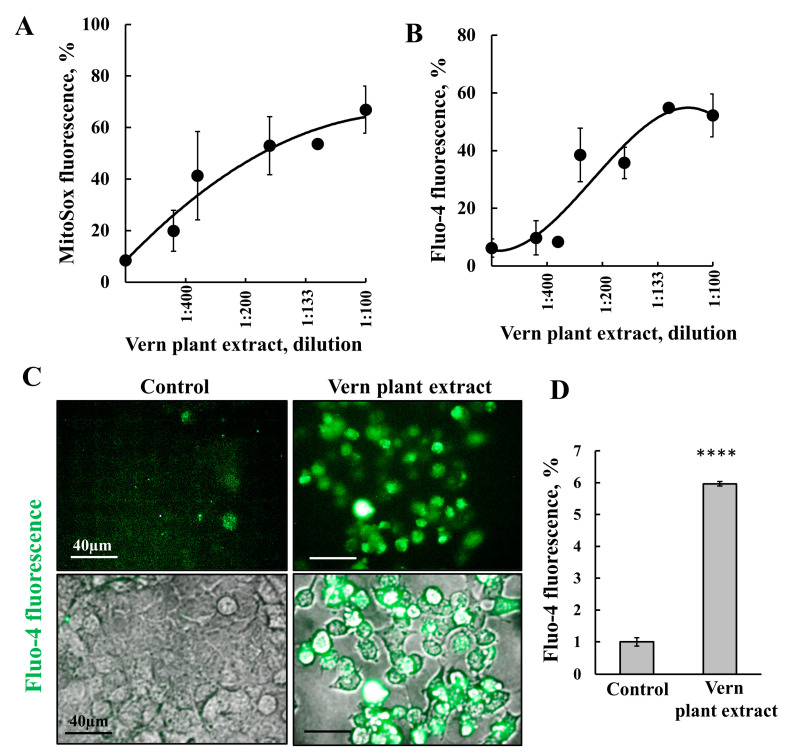
**Vern plant extract induced increased intracellular Ca^2+^ and ROS production**. SH-SY5Y cells were incubated for 24 h with the indicated dilutions of Vern plant extract and analyzed for ROS production using MitoSox Red reagent and FACS analysis (**A**), and for intracellular [Ca^2+^]i, using the calcium indicator Fluo-4 reagent and FACS analysis [46] (**B**), or for Vern plant extract treatment at 1:333 dilution is visualized using Operetta imaging (**C**), and quantified (**D**) **** *p* < 0.0001.

**Figure 4 cancers-15-01627-f004:**
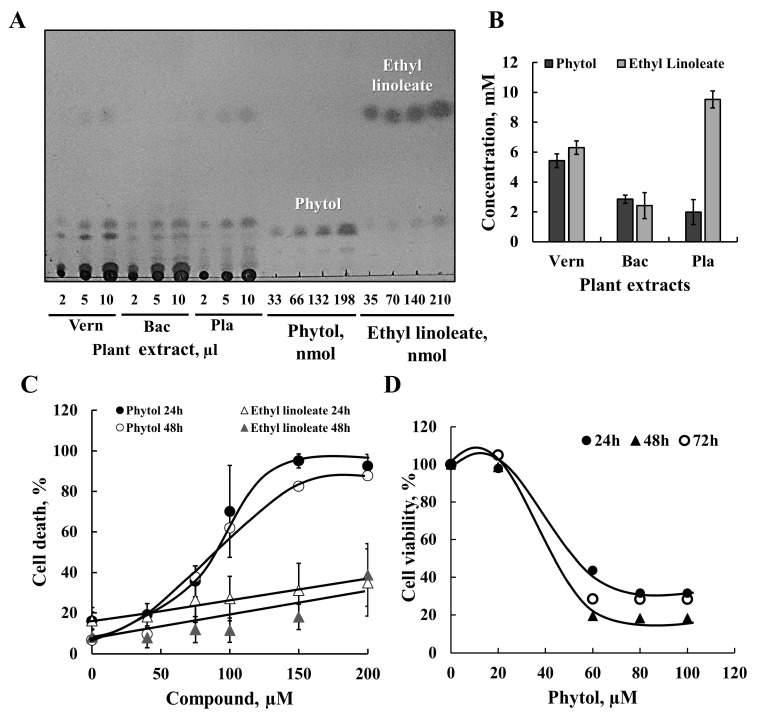
**GC-MS analysis of the plant extracts and phytol as one of the cell death active compounds**. (**A**) Ethanolic extracts from Vern, Bac, and Pla plant extracts were subjected to TLC separation, along with known amounts of phytol and ethyl linoleate using the solvent mixture of petroleum ether:diethyl ether:acetic acid (85:15:1, V:V:V) and developed by exposure to iodine vapor. (**B**) Quantification of phytol and ethyl linoleate in plant extracts using the compounds’ calibration curves and Image J software (*n* = 3). (**C**) SH-SY5Y cells were incubated for 24 h or 48 h (**C**) with the indicated concentration of phytol or ethyl linoleate, and cell death was evaluated using PI staining and FACS analysis. Results are the mean ± SEM (*n* = 3). (**D**) Cell survival was assayed following 24, 48, or 72 h incubation with the indicated concentrations of phytol, using an XTT assay.

**Figure 5 cancers-15-01627-f005:**
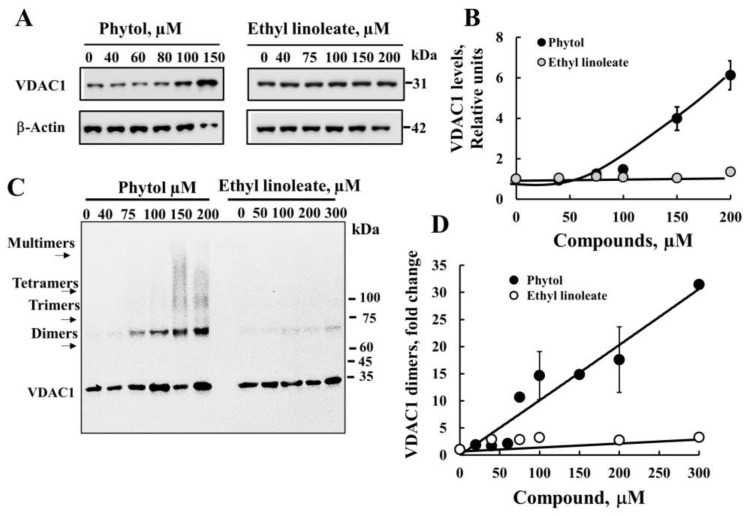
**Phytol and ethyl linoleate VDAC1 overexpression and oligomerization.** (**A**,**B**) SH-SY5Y cells were incubated for 24 h with the indicated concentrations of phytol or ethyl linoleate, and then analyzed for VDAC1 expression levels by immunoblotting (**A**). The level of VDAC1 is given as relative units (RUs) quantified using Image J software (**B**). (**C**,**D**) Control and phytol- or ethyl linoleate-treated cells (1 mg/mL) were also analyzed for VDAC1 oligomerization by incubation with the cross-linking reagent EGS (100 μM), followed by immunoblotting using anti-VDAC1 antibodies (**C**) and quantified for dimer level (**D**). The position of the VDAC1 monomers, dimers, trimers, and tetramers is indicated. Results are the mean ± SEM (*n* = 3).

**Figure 6 cancers-15-01627-f006:**
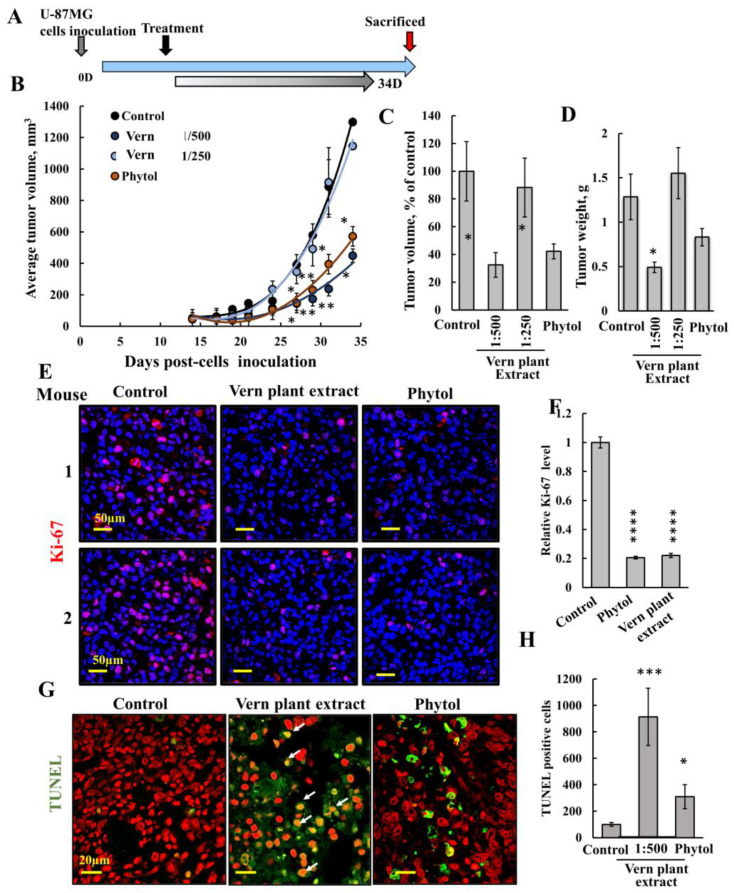
**Inhibition of tumor growth by Vern plant extract and phytol in a GBM xenograft mouse model.** U-87MG cells (1.8 × 10^6^ cells/mouse) were s.c. inoculated into nude mice. Tumor volume was monitored (using a digital caliper), and on day 14, when the tumor volume was about 50 mm^3^, the mice were divided into four groups with a similar average volume calculated per group (5 or 6 mice per group) (**A**). The four mice groups were subjected to the following treatments: control (ethanol to a final concentration 0.14%) or Vern plant extract to a final dilution of 1:250 or 1:500, and phytol to a final 75 μM, calculated according to the tumor volume. (**B**) The calculated average tumor volumes as a function of time are presented as means ± SEM (*n* = 5 or 6 mice). (**C**) Tumor-calculated volume before scarifying the mice (day 34), presented as % of the control. (**D**) The calculated average tumor weights are presented as means ± SEM. * *p* < 0.05. (**E**,**F**) Confocal images of representative paraffin-embedded sections from U-87MG-derived control, Vern plant extract (1:500)-, or phytol (75 μM)-treated tumors, immunofluorescent-stained with antibodies against the proliferation marker, Ki-67 (**E**), and quantification of staining intensity (**F**). Results are the mean ± SEM (*n* = 3), **** *p* < 0.0001. (**G**,**H**) TUNEL staining on paraffin-embedded sections cut from control, Vern plant extract-, or phytol-treated tumors. TUNNEL staining of tumor sections was carried out as described in the Section 2. Representative confocal images with red staining indicates PI nuclear staining, and green-stained cells indicate TUNEL staining (**G**). TUNEL-positive cells were quantified and presented as TUNEL-positive cells (**H**). Results are the mean ± SEM (*n* = 3), * *p* < 0.05, *** *p* < 0.001.

**Figure 7 cancers-15-01627-f007:**
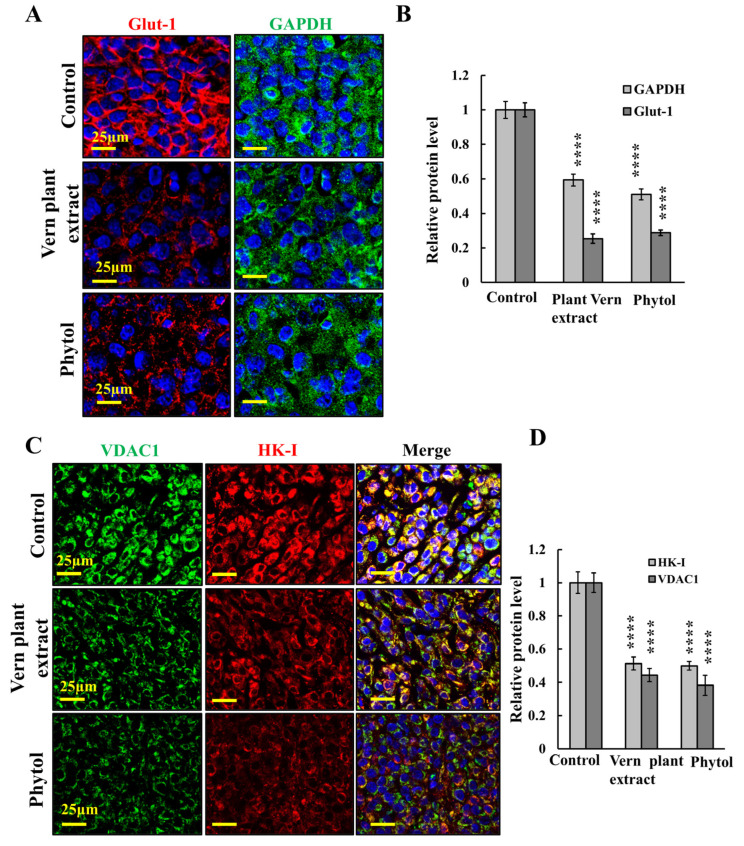
**Tumor treatment with Vern plant extract or phytol resulted in reduced cell metabolism.** (**A**–**D**) Confocal images of sections from U-87MG-derived tumors, control or treated with Vern plant extract (1:500) or phytol (75 μM), immunofluorescent-stained for Glut-1 and GAPDH (**A,B**) or for VDAC1 and HK-I (**C**,**D**) using specific antibodies. Staining intensity was quantitative using Image J software (**B**,**D**). Results reflect the mean ± SEM (*n* = 3), **** *p* ≤ 0.0001.

**Figure 8 cancers-15-01627-f008:**
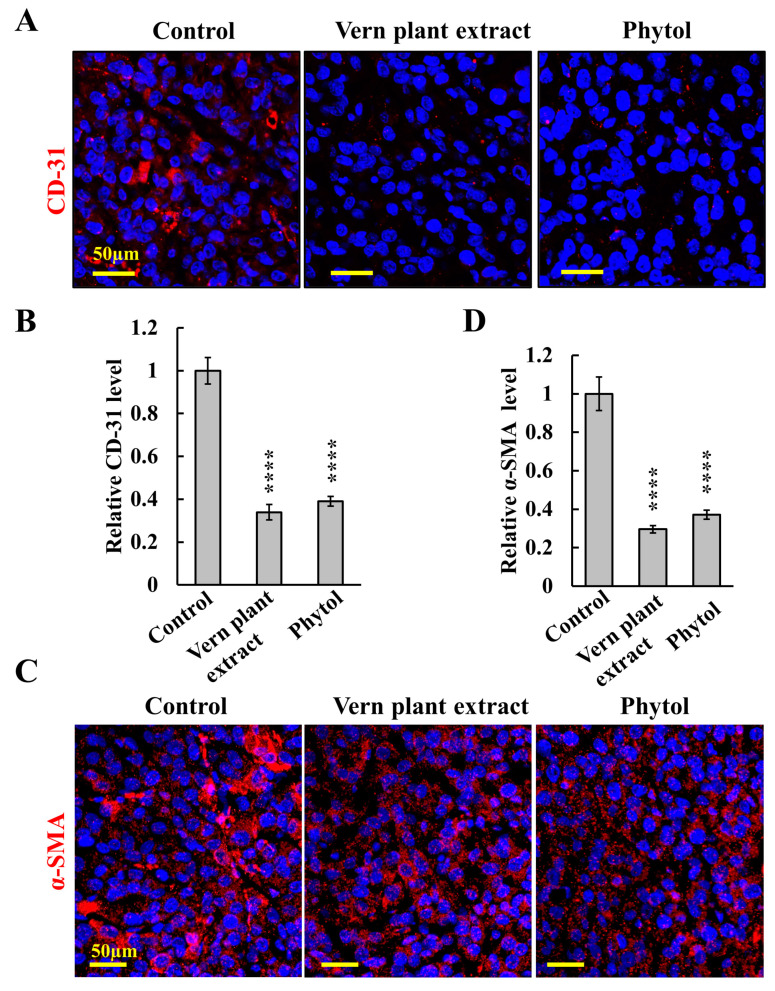
**Vern plant extract and phytol treatment markedly reduced angiogenesis and the tumor microenvironment in U-87MG cell-derived tumors**. Confocal images of sections from U-87MG-derived tumors, control or Vern plant extract (1:500) or phytol (75 μM)-treated tumors were immunofluorescent-stained for CD-31 (**A**,**B**) or α-SMA (**C**,**D**), and their quantification is presented (**B**,**D**). Results are means ± SEM (*n* = 3 mice), **** *p* ≤ 0.0001.

**Figure 9 cancers-15-01627-f009:**
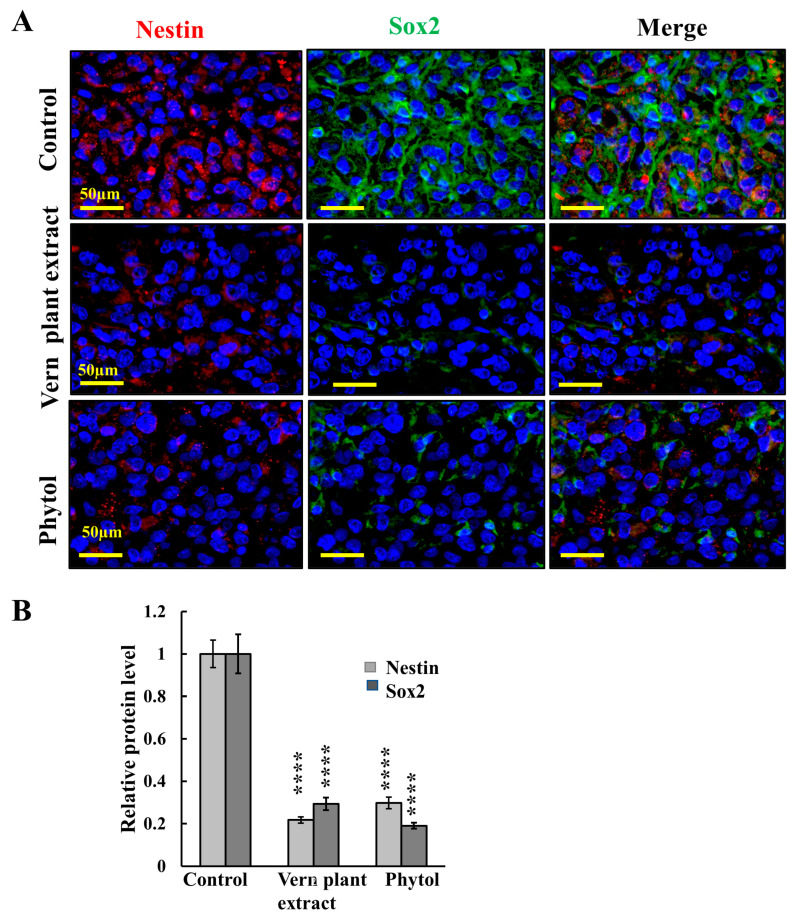
**Vern plant extract and phytol treatment markedly reduced cancer stem cells markers in U-87MG cell-derived tumors.** Representative IF staining of tumor sections from U-87MG-derived tumors, control or treated with Vern plant extract (1:500) or phytol (75 μM), co-immunofluorescent-stained with specific antibodies against the CSCs markers, Sox2 and Nestin (**A**), and their quantitative analysis (**B**). Results are means ± SEM (*n* = 3 tumors), **** *p* ≤ 0.0001.

**Figure 10 cancers-15-01627-f010:**
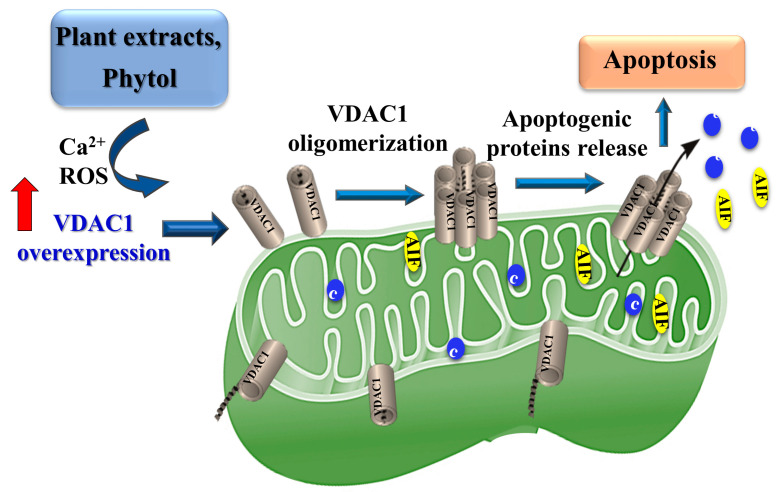
Proposed model for active molecules of plant extracts inducing VDAC1 overexpression and VDAC1 oligomerization leading to apoptosis.

## Data Availability

Not applicable.

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
