# Peer review of "Pro-Apoptotic and Anti-Cancer Activity of the Vernonanthura Nudiflora Hydroethanolic Extract"

_cancers, 2023, doi:10.3390/cancers15051627_

Round 1
Reviewer 1 Report
The article was well written and informative. The only change is to remove the first paragraph of the introduction because it is repeated by the second paragraph.
Author Response
We thank this reviewer for the comment. We have removed the paragraph in the Introduction to eliminate duplication.
Reviewer 2 Report
In this paper, authors revealed that extracts derived from Vernonanthura nudiflora show anti-cancer effect by inducing apoptosis. In detailed mechanism study, author found this extract induce overexpression and oligomerization of VDAC1. By analyzing the Vern extract components by GC-MS, authors found phytol is one of major compound that induced apoptosis. In the end, authors also validated Vern extract’s anti-cancer effect in vivo. However, the data in animal model is opposite to result in vitro. As VDAC1 is decreased upon the extract treatment in the mouse model.
1. Figure1, Whether the extract induce cytotoxicity in normal cell as well?
2. Figure1C, can author show the flow figure of apoptosis assay in supplementary data.
3. Figure 2, except glioblastoma and neuroblastoma, whether Vern extract also induce cytotoxicity in other major cancer types, such as colon cancer, liver cancer and ovarian cancer.
4. Figure3, as VDAC1 is a mitochondrial membrane channel protein, authors should detect if the treatment affect Mitochondrial membrane potential here.
5. Figure 7C, why VDAC1 is decreased upon phytol treatment in vivo, while it is up-regulated in vitro upon phytol treatment.
Author Response
Reviewer 2
We thank this reviewer for the comments that were addressed in the revised version of the MS.
- Figure1, Whether the extract induce cytotoxicity in normal cell as well?
To address this question, we compared the effects of the Vern extract on the non-cancerous cells’ MEFs and the cell line used in most of the presented study, SHYH5. The results were added as new Fig. S2B and indicate that the non-cancerous cells are less sensitive than the cancerous ones.
- Figure1C, can author show the flow figure of apoptosis assay in supplementary data
As suggested, we added a representative FACS analysis of Annexin V/PI staining as Fig. S1.
- Figure 2, except glioblastoma and neuroblastoma, whether Vern extract also induce cytotoxicity in other major cancer types, such as colon cancer, liver cancer and ovarian cancer.
The Vern plant extract was tested in other cell lines as prostate PC-3 and Hela cells. This is now added to the supplementary data (Fig. S2A).
- Figure 3, as VDAC1 is a mitochondrial membrane channel protein, authors should detect if the treatment affect mitochondrial membrane potential here.
It has been reported that in some apoptotic systems, loss of Δψm may be an early event in the apoptotic process. However, there are emerging data suggesting that, depending on the model of apoptosis, the loss of Δψm may not be an early requirement for apoptosis, but may actually be a consequence of the apoptotic-signaling pathway [J. D. Ly, D. R. Grubb, & A. Lawen. The mitochondrial membrane potential (Δψm) in apoptosis; an update. Apoptosis vol. 8, pp. 115–128 (2003)].
VDAC1 is an outer mitochondrial membrane (OMM) and the apoptotic proteins to be released are in the intramitochondrial membrane space; thus, their release requires permeability changes only in the OMM. We carried out this experiment previously and repeated it again with the same results (Fig. S3). Only when high (over 80%) cell death was obtained was the Dy decreased. This suggests that at high levels of cell death, the cells were distracted including the mitochondria.
We added the following to the Results section:
Cell treatment with Vern plant extract reduced the mitochondrial membrane potential (Dy) only when high (over 80%) cell death was obtained (Fig. S3).
The findings that the increase in ROS production, [Ca2+] levels, and dissipation of (Dy) were not correlated with cell death and were observed only at high concentrations of the Vern plant extract and over 80% cell death suggest that these effects are due to cell destruction including of the mitochondria.
Figure 7C, why VDAC1 is decreased upon phytol treatment in vivo, while it is up-regulated in vitro upon phytol treatment.
This is an interesting question, but we have seen this in many of our studies with several cancer mouse models (see new Ref. [92]). We explain this in the following: In the in-vitro experiments conducted for 24 or 48 h, the stress conditions induced VDAC1 overexpression, resulting in apoptosis (see Fig. 5 in [92]). In the in-vivo study, the tumors were treated for 27–45 days (depending on the treatment type. In the Vern plant extract or Phytol treatment for 27 days, the obtained massive cell death in the tumors resulted in cell distraction, including in the mitochondria and in degradation of many cell proteins.
We have indicated this possible explanation in the Discussion: “The decrease in the expression of metabolism-related enzymes in the tumors treated with Vern plant extract or Phytol for 27 days may result from the massive cell death leading to cell distraction, including in the mitochondria and in degradation of many cell proteins (Fig. 7). Similar results were obtained with the VDAC1-based peptide [92].”
Reviewer 3 Report
The manuscript is well prepared, well written, well presented and the overall quality is good
There are no major comments on this work, the experiments are well designed and the results are clear and presented well.
Author Response
We thank this reviewer for the positive comments
Round 2
Reviewer 2 Report
Authors addressed my concerns properly, I do not have further questions, I think this paper can be accepted for publication now.